# Renal Actinomycosis in Humans—A Narrative Review

**DOI:** 10.3390/microorganisms12091922

**Published:** 2024-09-21

**Authors:** Ilias Giannakodimos, Afroditi Ziogou, Alexios Giannakodimos, Evangelia Mitakidi, Aris Kaltsas, Zisis Kratiras, Michael Chrisofos

**Affiliations:** 1Third Department of Urology, Attikon University Hospital, School of Medicine, National and Kapodistrian University of Athens, 124 62 Athens, Greece; iliasgiannakodimos@gmail.com (I.G.); ares-kaltsas@hotmail.com (A.K.); kratiras.urology@gmail.com (Z.K.); 2Department of Medical Oncology, Metaxa Cancer Hospital, 185 37 Athens, Greece; aziogou@yahoo.com (A.Z.); alexisgiannak@hotmail.com (A.G.); 3Department of Anesthesiology, General Hospital of KAT, 145 61 Athens, Greece; evangeliamitakidi@gmail.com

**Keywords:** actinomycosis, renal, upper urinary tract

## Abstract

Actinomycosis of the kidney is extremely rare, with only a few cases reported in the literature. This rare entity usually presents with common clinical manifestations and non-specific imaging findings, thus rendering its diagnosis extremely challenging. According to case reports published in the literature, fever was present in the majority of cases (56.25%). Several risk factors have been related to the development of renal actinomycosis, including a history of urogenital surgery, urolithiasis, or urogenital cancer. Initial diagnostic investigation consists of abdominal ultrasonography (15 patients, 46.88%) and abdominal computed tomography (23 patients, 48.15%). Regarding therapeutic approach, 5 patients (16.67%) received only antibiotic treatment, 5 patients (16.67%) underwent surgery, and 20 patients (66.67%) received both antibiotic and surgical treatment. Accurate diagnosis relies on the clinician’s high index of suspicion and is ultimately confirmed through histological examination or cultures, obtained either preoperatively via biopsies or postoperatively after surgical removal of the infected kidney. To our knowledge, this is the first narrative review of the literature that collects knowledge concerning infection of UUT induced by dysbiosis of actinomycosis species. The aim of this narrative review was to systematically search the literature on primary renal actinomycosis, highlighting the diagnostic approach and treatment strategies for the management of this rare entity.

## 1. Introduction

Actinomycosis is a rare infection provoked by anaerobic Gram-positive bacteria of the Actinomyces genus [1]. These bacteria typically inhabit the human oral cavity, digestive system, and urinary tract (UT). Due to their low virulence, colonization of the UT from these bacteria seldom causes active infections in individuals with normal immune function [2]. However, the incidence of actinomycosis increases in immunosuppressed individuals, diabetic patients, or those with mucosal damage from surgery or trauma [1,2]. Their infection most commonly affects the cervicofacial area, abdominopelvic region, and respiratory tract [1]. Interestingly, actinomycosis of the kidney is extremely rare, with only a few cases documented in the medical literature [3]. This benign entity usually presents with vague symptoms and non-specific imaging findings, thus rendering its diagnosis extremely challenging [4]. Accurate diagnosis relies on the clinician’s high index of suspicion and is ultimately confirmed through histological examination or cultures, obtained either preoperatively via biopsies or postoperatively after surgical removal of the infected kidney [5]. Renal actinomycosis is frequently misdiagnosed, especially due to its atypical clinical presentation, resembling other conditions such as malignancy, tuberculosis, and nocardiosis [6,7]. Treatment options include extended antibiotic therapy or surgical removal of the infected kidney [3,8]. This study aims to systematically review the literature on primary renal actinomycosis, emphasizing on diagnostic methods and treatment strategies of choice for this uncommon condition.

## 2. Materials and Methods

This narrative review was performed to export data on actinomycosis species UUT infections in humans. The literature research for this narrative review was performed according to the Preferred Reporting Items for Systematic Reviews and Meta-Analyses guidelines [9]. Two investigators (A.Z., A.G.) independently searched PubMed/Medline and Scopus databases for potential articles reporting on actinomycosis of the kidney until 23 August 2024. The following keywords were used for the search strategy: “Actinomycosis” OR “Actinomyces” AND “kidney” OR “renal”. Any dispute was resolved by the intervention of a senior investigator (I.G.). This study included case reports and case series of Actinomyces spp. infections deriving from the kidney in males and females of all races and ages, published from 1924 to 2022. The inclusion criteria mandated that all eligible studies be exclusively written in the English language. Only articles presenting primary data, such as case reports or case series, were accepted; reviews, systematic reviews, and retrospective studies were excluded. Moreover, articles reporting no original or inadequate data, animal studies, comments, and publications designated as “epub ahead of print” were omitted from consideration. Articles deemed inaccessible were also not included. Furthermore, an additional search was conducted through the references of the included articles according to the snowball procedure to identify additional eligible studies. 

The data extraction process was performed autonomously by three independent investigators (A.Z., E.M., A.G.). Data from all cases included in the present review was systematically collected using a pre-defined template. Information regarding age, epidemiology, symptoms, and patients’ medical history was thoroughly gathered. Additionally, details concerning diagnostic modalities, therapeutic approach, and clinical outcome of renal actinomycosis in humans were accumulated by the authors and used for the extraction of results. 

## 3. Results

### 3.1. Literature Search

Literature search yielded a total of 845 studies. Subsequent to the elimination of duplicate studies, record screening, and the implementation of the snowball procedure, a total of 32 articles, published from 1924 to 2022, met the inclusion criteria and were included in this systematic review. The flow chart presenting the systematic search of this narrative review is shown in Figure 1.

### 3.2. Demographic Characteristics

In total, the included articles concerned 32 patients, 18 males (56.25%) and 14 (43.75%) females, with a male/female ratio of 1.3. Mean age of included patients was 43.45 ± 20.80 (mean, SD), varying from 6 to 80 years. A total of 11 cases (40.74%) originated from European countries, 10 cases (37.04%) from Asian countries, 5 cases (18.52%) from the USA, and only one case (3.7%) from Australia. Regarding potential risk factors associated with the development of renal actinomycosis, four patients (12.50%) had undergone urogenital surgery, three patients (9.38%) presented with history of urolithiasis, and one patient (14.81%) had a history of urogenital cancer (3.13%), while no patients presented with a history of hydronephrosis or immunodeficiency.

### 3.3. Clinical Manifestations

Concerning clinical manifestations of the included patients, 93.75% complained of various symptoms, while only two patients (6.25%) remained asymptomatic. Fever was present at the majority of cases (18 patients, 56.25%), and 16 patients (50%) reported weight loss, 14 patients (43.75%) flank pain, and 13 patients (40.63%) abdominal pain. Furthermore, six patients (18.75%) complained of anorexia, five patients (15.63%) reported hematuria, and four patients (12.50%) reported symptoms related with pyelonephritis, while abdominal swelling, nausea, and vomiting comprised infrequent clinical manifestations. A detailed symptomatology of the included patients is shown in Table 1. Out of the available data, mean duration of symptoms was estimated at 8.33 ± 25.73 months (mean, SD). The maximum duration of symptoms was reported as ten years, while acute onset of symptoms in less than one week was recorded in two (9.52%) cases.

### 3.4. Diagnostic Approach

Concerning the diagnostic approach of this rare entity, abdominal ultrasonography (US) was performed in 15 patients (46.88%), abdominal computed tomography (CT) in 23 patients (48.15%), intravenous pyelography in 7 patients (25%), and abdominal magnetic resonance imaging (MRI) in 4 patients (12.50%). Additionally, antegrade pyelography, retrograde pyelography, and PET-CT were performed in one, three, and one patients, respectively. Imaging findings of the included cases are summarized in Table 2.

Out of the 32 included cases, only 5 cases (15.63%) were diagnosed preoperatively. Preoperatively, urine cultures were obtained in 10 patients (34.48%), blood cultures in 4 patients (14.29%), renal abscess cultures in 5 patients (17.24%), and renal biopsy in 1 patient (3.13%). Interestingly, only renal abscess cultures were positive on all punctured patients (100%), while blood and urine cultures failed to identify any Actinomyces species. Initial diagnosis was misinterpreted and required further investigation in 18 cases (56.25%). Finally, out of the 32 cases, ultimate diagnosis was postoperative in 27 patients (%), either via histologic findings of excised specimens (21 cases, 65.63%), via cultures obtained during surgery (2 cases, 6.25%), or via both culture and histologic findings (4 cases, 12.50%).

### 3.5. Therapeutic Approach

Regarding therapeutic approach, out of the available data, 5 patients (16.67%) received only antibiotic treatment, 5 patients (16.67%) underwent surgery, and 20 patients (66.67%) received both antibiotic and surgical treatment. More specifically, ceftriaxone was administered in four patients (14.83%), cefuroxime in two patients (7.41%), doxycycline in three patients (11.1%), penicillin in two patients (7.41%), metronidazole in two patients (7.41%), amikacin in two patients (7.41%), and amoxicillin/clavulanic acid, ciprofloxacin, vancomycin, azithromycin, and meropenem only in one patient, respectively (3.7%). Type of surgery and antibiotic treatment for patients included in this comprehensive review are shown in Table 3. Concerning surgical approach, 20 patients (76.92%) underwent radical nephrectomy, 3 patients (11.54%) partial nephrectomy, 1 patient (3.85%) ureteral stent placement, 1 patient (3.85%) open surgery for debriding and drainage of pus, and 1 patient (3.85%) exploratory laparotomy and perirenal fat removal. Interestingly, in four patients (12.5%) a percutaneous nephrostomy tube or puncture was placed before surgery. Out of the surgically treated patients, 12 patients (40%) received antibiotic therapy preoperatively. In total, only one patient (3.57%) died due to renal actinomycosis in the first month after ureteral stent placement.

## 4. Discussion

Actinomycosis is a rare, chronic granulomatous disease provoked by infection with Actinomyces species [10]. These bacteria are part of the Actinomyceta class, characterized as Gram-positive, filamentous, facultative anaerobic and non-spore-forming bacilli [11]. Actinomyces is commonly found as part of the normal human microbiome, predominantly residing in the oropharynx, urogenital tract, and gastrointestinal tract, and typically presents low pathogenicity under normal conditions [12]. The Actinomyces species most frequently associated with human infections include Actinomyces israelii, Actinomyces naeslundii, Actinomyces viscosus, Actinomyces odontolyticus, Actinomyces gerencseriae, and Actinomyces meyeri [11]. Lesions caused by these opportunistic pathogens often occur in immunocompromised patients or following the disruption of the mucosal barrier, which allows the bacteria to invade directly [10]. The cervicofacial area is the most common site of infection, accounting for approximately 50% of cases, followed by the abdominopelvic area (20%) and pulmonary actinomycosis [13]. Regarding the urinary tract, and especially the kidneys, actinomycosis is exceedingly rare [10]. In general, anaerobic bacteria are not involved in kidney infections; conditions such as urinary obstruction, renal transplants, or necrotic tumors may lead to anaerobic bacteria proliferation. In that way, Actinomyces inhabits the urothelium by penetration through damaged epithelium and can also invade across several tissue planes resulting in the formation of masses and fistulous tracts [14,15]. Although uncommon, hematogenous spread to the kidney may also be suspected [16]. The rarity and severity of renal actinomycosis underlines the significance of this systematic review, which, to our knowledge, is the first comprehensive analysis focusing on actinomycosis located in the kidney.

Actinomycosis is frequently linked to immunosuppression resulting from conditions like malignancies, HIV, chemotherapy, or organ transplants [17]. Older age and male gender are also recognized as potential risk factors for the development of the disease; the present review noted that 56.25% of cases were male, while only one patient had a history of urogenital cancer (3.13%) [17]. Additionally, individuals with multiple comorbidities, such as urolithiasis or hydronephrosis, long-term medication use, malnutrition, or a history of abdominal surgery are exposed to a higher infection risk [18,19,20]. Specifically, in this comprehensive review, four patients (12.50%) had undergone urogenital surgery and three patients (9.38%) presented with a history of urolithiasis. Interestingly, no patient presented with history of hydronephrosis or immunodeficiency. Foreign objects, such as intrauterine devices (IUDs), urinary catheters, and nephrostomy tubes, may also contribute to infection by causing mucosal injuries and facilitating bacterial infiltration [20]. Nonetheless, in most cases of this study, no predisposing factors were detected, leaving the infection’s pathogenesis unclear. 

The clinical presentation of renal actinomycosis may often be non-specific. Fever is the most common initial symptom, while some patients may also experience weight loss and flank or abdominal pain [5]. In this study, fever was observed in 18 patients (56.25%), weight loss in 16 patients (50%), flank pain in 14 patients (43.75%), and abdominal pain in 13 patients (40.63%). Hematuria has also been reported, and was present in five patients (15.63%) in our study. Additionally, other non-specific manifestations such as anorexia, pyelonephritis-related symptoms, abdominal swelling, nausea, and vomiting were sporadically noted [16,21]. Typically, these symptoms persist for prolonged periods of time and acute onset of the infection is rarely observed; in accordance with these findings, acute onset of symptoms, in less than one week, was reported in only two (9.52%) cases in our study. Given its atypical and evolving clinical presentation, actinomycosis of the kidney can resemble malignancies or various other infections [5,7]. Effective treatment requires early clinical suspicion; therefore, physicians should be suspicious about this rare condition and include it in their differential diagnosis. 

The diagnostic approach to renal actinomycosis remains challenging, since regular laboratory examinations and imaging techniques are not disease-specific. Although kidney ultrasound is commonly performed, findings remain ambiguous and mainly depict the presence of renal masses [19]. A significant number of patients (48.15%) reported in the literature were subjected to a CT scan; findings consisted of infiltrative cystic masses or masses with central necrosis located in the kidney, enlargement of the surrounding lymph nodes, such as the retroperitoneal lymph nodes, as well as fluid collection in the abdominal cavity [5,20]. Fistulas and involvement of several anatomic structures were also observed [4,5]. Moreover, other imaging modalities, such as MRI, intravenous pyelography, or even PET-CT, have been used to aid in the diagnosis of renal actinomycosis in the literature. These imprecise imaging features may easily lead to misdiagnosis; tumors such as adenocarcinomas, as well as renal abscesses, cysts, or other infections, including tuberculosis or Proteus infection of the kidney, should be included in the differential diagnosis [4,7,19]. Notably, high misdiagnosis rates are also described in our comprehensive analysis; misinterpretation of the underlying disease was observed in 56.25% of the included patients, highlighting the need for further investigation and more specific diagnostic tools [7]. 

Preoperative diagnosis is particularly demanding and can only be established by isolation of Actinomyces spp. in cultures or histological examination of the infected renal tissue. This difficulty is underlined by the low percentage of preoperative diagnoses in our review (15.63%). Among the included patients, cultures of blood, urine, infected renal tissue, and abscess drainage material have been performed preoperatively; however, only renal abscess cultures were positive in all punctured patients, while the presence of bacterial growth in blood or urine may not be detected [6,20]. Significantly, Actinomyces infection might be polymicrobial; Gram-positive bacteria, such as Enterococcus spp. and Staphylococcus spp., or Gram-negative bacteria, including Proteus spp., have been isolated from cultures of patients with renal actinomycosis [20,22]. The existence of other microorganisms may inhibit Actinomyces culture growth. Moreover, culture results depend on previous antibiotic administration and might remain sterile in the case of recent antimicrobial treatment [23]. Another limitation when performing cultures is the prolonged period of up to 20 days required to detect bacterial proliferation. In cases where actinomycosis is suspected, a CT- or US-guided fine-needle aspiration of the liquefied central portions of the mass should be performed to confirm the diagnosis histologically [5]. This method has only been successfully used on a limited basis so far [5]. Histological examination supports and facilitates the diagnostic approach; it is conducted either preoperatively from material retrieved by biopsies or postoperatively on the resected surgical specimen. Microscopically, infection by Actinomyces spp. presents with formation of chronic granulomas, areas with fibrosis, and groups of Gram-positive filamentous bacteria with clubbed appendages composing the typical “sulfur granules” [24,25]. Due to the heterogeneity on reported histologic findings in the present review, no further analysis could be performed. The typical “sulfur granules” are histological characteristics for actinomycosis infection, observed in about 75% of cases, and constitute a mechanism of resistance to phagocytosis, thus enabling bacterial proliferation [1]. Additionally, Gram staining reveals typical Gram-positive filamentous bacilli peripheral to the sulfur granules and shows higher sensitivity than cultures in regards to pathogen detection [1].

Due to difficulties in the diagnostic approach of this rare entity, surgery is widely considered both a diagnostic and treatment option in the majority of cases. Notably, in our review, 84.4% of the included patients were diagnosed postoperatively. Surgery in renal actinomycosis mainly consists of radical resection of the infected kidney, and occasionally closure of any existing fistulas and abscess drainage [6,24]. Surrounding infected tissue removal may also be required. In the majority of cases, surgical excision of the infected tissues is related to favorable clinical outcome [26]. Surgery is also considered the optimal choice in more complex cases, such as fistula formation, in which antibiotic therapy does not comprise an effective therapeutic option [4]. According to our study, the majority of included patients (76.92%) underwent radical nephrectomy, while 11.54% of patients were subjected to partial nephrectomy, 3.85% to ureteral stent placement, 3.85% to open surgery for debriding and drainage of pus, and 3.85% to exploratory laparotomy and perirenal fat removal. More specifically, in specific patients (12.5%), a percutaneous nephrostomy tube or puncture can be placed before surgery, but there are no specific indications (only in 12.5% of our patients). A study by Valfour et al. examined the potency of antibiotics in multiple cases of actinomycosis infection and suggested extended antibiotic regimens with beta lactams, consisting of penicillin G or amoxicillin, for 6–12 months accompanied by surgical excision in more complicated cases [1]. Thus, in cases of early clinical suspicion, conservative treatment with high doses of intravenous penicillin constitutes an accepted alternative solution and surgery can be avoided [19,27]. Ampicillin, tetracyclines, aminoglycosides, or cephalosporins are alternative antimicrobial agents to treat infection by Actinomycoses species and have been provided to patients included in this review; however, they are not widely administered [4,20]. In order to prevent further disease expansion, antibiotics may be given preoperatively or postoperatively; intriguingly, intraoperative administration has also been described in the literature with a favorable clinical outcome [4]. According to this comprehensive review, 5 patients (16.67%) received only antibiotic treatment, 5 patients (16.67%) underwent surgery, and 20 patients (66.67%) received both antibiotic and surgical treatment. More specifically, ceftriaxone was administered in four patients (14.83%), cefuroxime in two patients (7.41%), doxycycline in three patients (11.1%), penicillin in two patients (7.41%), metronidazole in two patients (7.41%), amikacin in two patients (7.41%), and amoxicillin/clavulanic acid, ciprofloxacin, vancomycin, azithromycin, and meropenem in only one patient, respectively (3.7%). Interestingly, 66.7% of the included patients were subjected to both surgery and antibiotic administration and the majority secured an optimal clinical result. Early detection and effective treatment in renal actinomycosis generally led to a positive clinical outcome.

To the best of our knowledge, this is the first narrative review of the literature focusing on the epidemiology, clinical features, diagnostic processes, and therapeutic strategies for primary actinomycosis located in the kidney. However, this study presents with certain limitations. The review only included case series and case reports with sufficient data, whose credibility depends primarily on meticulous record-keeping. Additionally, the variability across institutions in surgical techniques and record-keeping practices significantly impacts outcomes and time-to-event analyses.

## 5. Conclusions

Renal actinomycosis is an extremely rare condition with diverse and often vague symptoms and imaging findings. Although it is uncommon, it should be considered in the differential diagnosis, particularly in immunocompromised patients or those who have had abdominal surgery and present with symptoms like fever, weight loss, abdominal or flank pain, and hematuria. Imaging techniques like CT or MRI are helpful for identifying the primary lesion. Renal biopsies obtained pre- or postoperatively are used for histological examination, which can confirm the presence of Actinomyces spp. Treatment typically involves antibiotics, with penicillin being the optimal choice, and may also require surgical removal of the infected areas. Effective management of renal actinomycosis relies on prompt clinical and microbiological recognition, early diagnosis, and appropriate antimicrobial therapy. Further case reports or retrospective studies are needed to further investigate this rare entity.

## Figures and Tables

**Figure 1 microorganisms-12-01922-f001:**
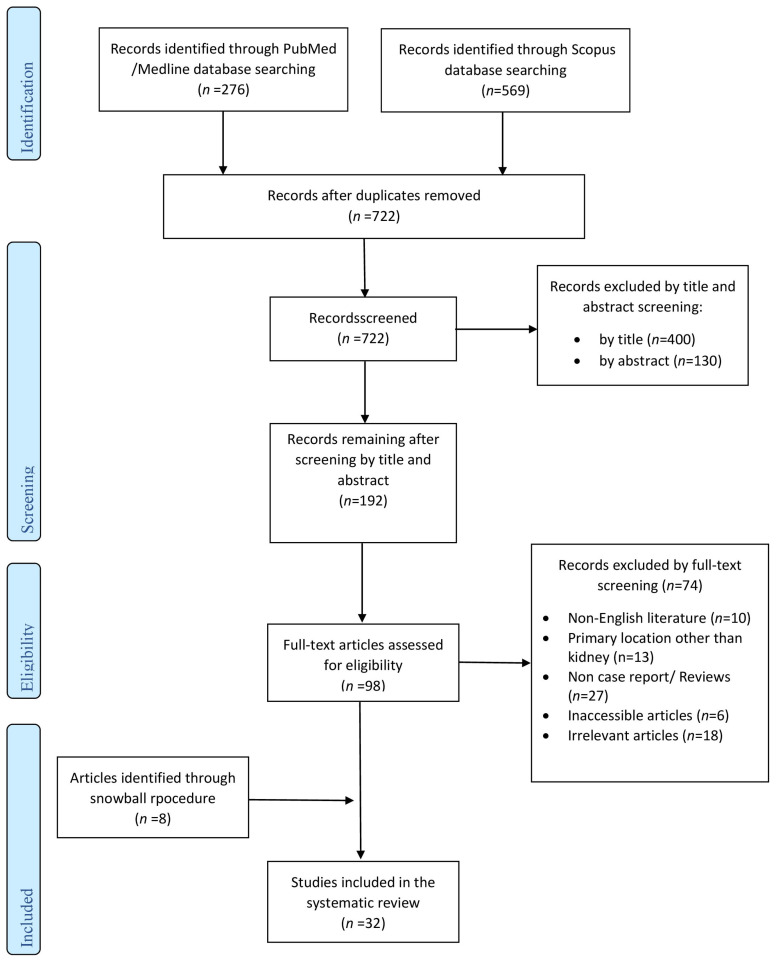
Flow-chart of the systematic search for this comprehensive review.

**Table 1 microorganisms-12-01922-t001:** Clinical manifestations of renal actinomycosis.

Symptom/Sign	Number of Patients	Percentage of Patients
Fever	18	56.25%
Weight loss	16	50%
Flank pain	14	43.75%
Abdominal pain	13	40.63%
Anorexia	6	18.75%
Hematuria	5	15.63%
Pyelonephritis-related symptoms	4	12.50%
Palpable mass	4	15.38%
Nausea	3	9.38%
Vomiting	3	9.38%
Abdominal swelling	2	6.25%
Asymptomatic	1	3.85%

**Table 2 microorganisms-12-01922-t002:** Imaging findings of gastric actinomycosis (NR: Not Reported).

Imaging Technique	Morphology	Size
Computed tomography	Gastric mass, irregular thickening of gastric wall, necrosis, invasion to adjacent tissues, lymphadenopathy	3–16 cm
Ultrasonography	Thickening of gastric wall, gastric mass	3–15 cm
Magnetic resonance imaging	Tumor-like lesion, infiltration to adjacent tissues, cystic areas	NR
Endoscopy	Subepithelial tumor-like lesions, yellowish exudate, ulceration, edema, necrosis, nodular areas	1.5–5 cm

**Table 3 microorganisms-12-01922-t003:** Type of surgery and antibiotic treatment in patients with renal actinomycosis.

Treatment	Type	Number of Patients (Percentage)	Total
Antibiotics	Ceftriaxone	4 (14.83%)	5 patients (16.67%)
	Cefuroxime	2 patients (7.41%)	
	Doxycycline	3 patients (11.1%)	
	Penicillin	2 patients (7.41%)	
	Metronidazole	2 patients (7.41%)	
	Amikacin	2 patients (7.41%)	
	Amoxicillin/clavulanic acid, ciprofloxacin, vancomycin, azithromycin, and meropenem	1 patient (3.7%)	
Surgery	Radical nephreactomy	20 patients (76.92%)	5 patients (16.67%)
	Partial nephrectomy	3 patients (11.54%)	
	Ureteral stent placement	1 patients (3.85%)	
	Open surgery for debriding and drainage of pus	1 patient (3.85%)	
	Exploratory laparotomy	1 patient (3.85%)	
Antibiotics and Surgery	-		20 patients (66.67%)

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
