# Peer review of "Renal Actinomycosis in Humans—A Narrative Review"

_microorganisms, 2024, doi:10.3390/microorganisms12091922_

Round 1

Reviewer 1 Report

Comments and Suggestions for Authors

Manuscript Renal actinomycosis in Humans- A narrative review by Giannakodimos et al. deals with reported case of mentioned diseases. In general, the paper is written nicely, and provides the complete review of the given topic. I found it quite interesting also because of my personal experience working with these microorganisms in laboratory research.

I have no extended comments/suggestions/questions for the Authors, except for the following: 1. In the Section Material and Methods, Authors could include whether there were limitation of date of publication of collected case reports. 2. Figure 2 should be technically arranged according to journal propositions. 3. Section results could be divided into sub-titles, in order for reader to navigate more easily through the section. 4. Section Discussion is written very nicely, it was quite interesting to read it. 5. Although it is not mandatory, a brief Conclusion would be very nice.

Best of wishes in publishing your paper!

Author Response

Comment 1:  In the Section Material and Methods, Authors could include whether there were limitation of date of publication of collected case reports. 

Reply: Thank you very much for your suggestion. The phrase “published from 1924 to 2022” has been added to Materials and Methods section to provide a timeframe of the included articles. No limitation of publication date was applied.

Comment 2: Figure 2 should be technically arranged according to journal propositions. 

Reply: Thank you for your comment. In our manuscript there is no Figure 2, so Reviewer 1 probably means Figure 1. Figure 1 has been technically arranged according to journal propositions and is resubmitted as a PNG file.

Comment 3: Section results could be divided into sub-titles, in order for reader to navigate more easily through the section.

Reply: Thank you for your suggestion. As requested, results section has been divided into sub-titles so as to facilitate the reader navigate through the section.

Comment 4: Section Discussion is written very nicely; it was quite interesting to read it. 

Reply: Thank you very much for your kind comments.

Comment 5: Although it is not mandatory, a brief Conclusion would be very nice.

Reply: Thank you for your helpful indication. Authors added a brief Conclusion after the Discussion section, as you proposed.

Reviewer 2 Report

Comments and Suggestions for Authors

In this narrative review, the Authors detailed the complex clinical picture described in literature renal actinomycosis in humans.

General comment:

This is an good review, providing a concise and useful summary of present knowledge of this rare infection. I have a few comments

Major comment

1) For completeness, I suggest adding one or two tables summarizing the treatments carried out (surgery, antibiotics, with details on the type of antibiotics and duration of therapy)

Comments on the Quality of English Language

good

Author Response

In addition, we recommend that you increase the text count of the manuscript
to 4000 (excluding references and author information).

Reply: Increase of text count has been made in the Discussion section according to journal's recommendation.

REVIEWER 2/ Comment 1: This is an good review, providing a concise and useful summary of present knowledge of this rare infection. I have a few comments. Major comment:  For completeness, I suggest adding one or two tables summarizing the treatments carried out (surgery, antibiotics, with details on the type of antibiotics and duration of therapy)

Reply: Thank you for your constructive comment. Authors added another table (Table 3) to summarize the treatments carried out in all included patients where data was available. The Table 3 is entitled “Type of surgery and antibiotic treatment in patients with renal actinomycosis”.

Round 2

Reviewer 2 Report

Comments and Suggestions for Authors

I have no further comments